# Robust Learning of Fixed-Structure Bayesian Networks

**Yu Cheng**
Department of Computer Science
Duke University
Durham, NC 27708
yucheng@cs.duke.edu

**Ilias Diakonikolas**
Department of Computer Science
University of Southern California
Los Angeles, CA 90089
ilias.diakonikolas@gmail.com

**Daniel M. Kane**
Department of Computer Science and Engineering
University of California, San Diego
La Jolla, CA 92093
dakane@ucsd.edu

**Alistair Stewart**
Department of Computer Science
University of Southern California
Los Angeles, CA 90089
stewart.al@gmail.com

## Abstract

We investigate the problem of learning Bayesian networks in a robust model where an $\epsilon$-fraction of the samples are adversarially corrupted. In this work, we study the fully observable discrete case where the structure of the network is given. Even in this basic setting, previous learning algorithms either run in exponential time or lose dimension-dependent factors in their error guarantees. We provide the first computationally efficient robust learning algorithm for this problem with dimension-independent error guarantees. Our algorithm has near-optimal sample complexity, runs in polynomial time, and achieves error that scales nearly-linearly with the fraction of adversarially corrupted samples. Finally, we show on both synthetic and semi-synthetic data that our algorithm performs well in practice.

## 1 Introduction

Probabilistic graphical models [KF09] provide an appealing and unifying formalism to succinctly represent structured high-dimensional distributions. The general problem of inference in graphical models is of fundamental importance and arises in many applications across several scientific disciplines, see [WJ08] and references therein. In this work, we study the problem of learning graphical models from data [Nea03, DSA11]. There are several variants of this general learning problem depending on: (i) the precise family of graphical models considered (e.g., directed, undirected), (ii) whether the data is fully or partially observable, and (iii) whether the structure of the underlying graph is known a priori or not (parameter estimation versus structure learning). This learning problem has been studied extensively along these axes during the past five decades, (see, e.g., [CL68, Das97, AKN06, WRL06, AHHK12, SW12, LW12, BMS13, BGS14, Bre15]) resulting in a beautiful theory and a collection of algorithms in various settings.

The main vulnerability of all these algorithmic techniques is that they crucially rely on the assumption that the samples are precisely generated by a graphical model in the given family. This simplifying assumption is inherent for known guarantees in the following sense: if there exists even a very small fraction of arbitrary outliers in the dataset, the performance of known algorithms can be totally compromised. It is important to explore the natural setting when the aforementioned assumption holds only in an approximate sense. Specifically, we study the following broad question:

**Question 1** (Robust Learning of Graphical Models). *Can we efficiently learn graphical models when a constant fraction of the samples are corrupted, or equivalently, when the model is slightly misspecified?*

In this paper, we focus on the model of corruptions considered in [DKK+16] (Definition 1) which generalizes many other existing models, including Huber's contamination model [Hub64]. Intuitively, given a set of good samples (from the true model), an adversary is allowed to inspect the samples before corrupting them, both by adding corrupted points and deleting good samples. In contrast, in Huber's model, the adversary is oblivious to the samples and is only allowed to add bad points.

We would like to design robust learning algorithms for Question 1 whose sample complexity, $N$, is close to the information-theoretic minimum, and whose computational complexity is polynomial in $N$. We emphasize that the crucial requirement is that the error guarantee of the algorithm is *independent* of the dimensionality $d$ of the problem.

## 1.1 Formal Setting and Our Results

In this work, we study Question 1 in the context of *Bayesian networks* [JN07]. We focus on the fully observable case when the underlying network is given. In the *non-robust* setting, this learning problem is straightforward: the "empirical estimator" (which coincides with the maximum likelihood estimator) is known to be sample and computationally efficient [Das97]. In sharp contrast, even this most basic regime is surprisingly challenging in the robust setting. For example, the very special case of robustly learning a Bernoulli product distribution (corresponding to an empty network with no edges) was analyzed only recently in [DKK+16].

To formally state our results, we first give a detailed description of the corruption model we study.

**Definition 1** ($\epsilon$-Corrupted Samples). *Given $0 < \epsilon < 1/2$ and a distribution family $\mathcal{P}$, the algorithm specifies some number of samples $N$, and $N$ samples $X_1, X_2, \ldots, X_N$ are drawn from some (unknown) ground-truth $P \in \mathcal{P}$. The adversary is allowed to inspect $P$ and the samples, and replaces $\epsilon N$ of them with arbitrary points. The set of $N$ points is then given to the algorithm. We say that a set of samples is $\epsilon$-corrupted if it is generated by this process.*

**Bayesian Networks.** Fix a directed acyclic graph, $G$, whose vertices are labelled $[d] \stackrel{\text{def}}{=} \{1, 2, \ldots, d\}$ in topological order (every edge points from a vertex with a smaller index to one with a larger index). We will denote by Parents($i$) the set of parents of node $i$ in $G$. A probability distribution $P$ on $\{0, 1\}^d$ is defined to be a *Bayesian network* (or *Bayes net*) with graph $G$ if for each $i \in [d]$, we have that $\Pr_{X \sim P}[X_i = 1 \mid X_1, \ldots, X_{i-1}]$ depends only on the values $X_j$ where $j \in$ Parents($i$). Such a distribution $P$ can be specified by its *conditional probability table*.

**Definition 2** (Conditional Probability Table of Bayesian Networks). *Let $P$ be a Bayesian network with graph $G$. Let $\Gamma$ be the set $\{(i, a) : i \in [d], a \in \{0, 1\}^{|\text{Parents}(i)|}\}$. Let $m = |\Gamma|$. For $(i, a) \in \Gamma$, the parental configuration $\Pi_{i,a}$ is defined to be the event that $X_{\text{Parents}(i)} = a$. The conditional probability table $p \in [0, 1]^m$ of $P$ is given by $p_{i,a} = \Pr_{X \sim P}[X_i = 1 \mid \Pi_{i,a}]$.*

Note that $P$ is determined by $G$ and $p$. We will frequently index $p$ as a vector. We use the notation $p_k$ and the associated events $\Pi_k$, where each $k \in [m]$ stands for an $(i, a) \in \Gamma$ lexicographically ordered.

**Our Results.** We give the first efficient robust learning algorithm for Bayesian networks with a known graph $G$. Our algorithm has information-theoretically near-optimal sample complexity, runs in time polynomial in the size of the input (the samples), and provides an error guarantee that scales near-linearly with the fraction of adversarially corrupted samples, under the following restrictions: First, we assume that each parental configuration is reasonably likely. Intuitively, this assumption seems necessary because we need to observe each configuration many times in order to learn the associated conditional probability to good accuracy. Second, we assume that each of the conditional probabilities is balanced, i.e., bounded away from 0 and 1. This assumption is needed for technical reasons. In particular, we need this to show that a good approximation to the conditional probability table implies that the corresponding Bayesian network is close in total variation distance.

Formally, we say that a Bayesian network is *c-balanced*, for some $c > 0$, if all coordinates of the corresponding conditional probability table are between $c$ and $1 - c$. Throughout the paper, we use

$m = \sum_{i=1}^{d} 2^{|\text{Parents}(i)|}$ for the size of the conditional probability table of $P$, and $\alpha$ for the minimum probability of parental configuration of $P$: $\alpha = \min_{(i,a) \in S} \Pr_P[\Pi_{i,a}]$. We now state our main result.

**Theorem 3** (Main). *Fix $0 < \epsilon < 1/2$. Let $P$ be a $c$-balanced Bayesian network on $\{0,1\}^d$ with known structure $G$. Assume $\alpha \geq \Omega(\epsilon\sqrt{\log(1/\epsilon)}/c)$. Let $S$ be an $\epsilon$-corrupted set of $N = \widetilde{\Omega}(m\log(1/\tau)/\epsilon^2)$ samples from $P$. [1] Given $G, \epsilon, \tau$, and $S$, we can compute a Bayesian network $Q$ such that, with probability at least $1 - \tau$, $d_{\mathrm{TV}}(P,Q) \leq \epsilon\sqrt{\ln(1/\epsilon)}/(\alpha c)$. Our algorithm runs in time $\widetilde{O}(Nd^2/\epsilon)$.*

Our algorithm is given in Section 3. We first note that the sample complexity of our algorithm is near-optimal for learning Bayesian networks with known structure. The following sample complexity lower bound holds even without corrupted samples:

**Fact 4** (Sample Complexity Lower Bound, [CDKS17]). *Let $\mathcal{BN}_{d,f}$ denote the family of Bernoulli Bayesian networks on $d$ variables such that every node has at most $f$ parents. The worst-case sample complexity of learning $\mathcal{BN}_{d,f}$, within total variation distance $\epsilon$ and with probability $9/10$, is $\Omega(2^f \cdot d/\epsilon^2)$ for all $f \leq d/2$ when the graph structure is known.*

Consider Bayes nets whose average in-degree is close to the maximum in-degree, that is, when $m = \Theta(2^f d)$, the sample complexity lower bound in Fact 4 becomes $\Omega(m/\epsilon^2)$, so our sample complexity is optimal up to polylogarithmic factors.

We remark that Theorem 3 is most useful when $c$ is a constant and the Bayesian network has bounded fan-in $f$. In this case, the condition on $\alpha$ follows from the $c$-balanced assumption: When both $c$ and $f$ are constants, $\alpha = \min_{(i,a) \in S} \Pr_P[\Pi_{i,a}] \geq c^f$ is also a constant, so the condition $c^f \geq \Omega(\epsilon\sqrt{\log(1/\epsilon)})$ automatically hold when $\epsilon$ is smaller than some constant. On the other hand, the problem of learning Bayesian networks is less interesting when the fan-in is too large. For example, if some node has $f = \omega(\log(d))$ parents, then the size of the conditional probability table is at least $2^f$, which is super-polynomial in the dimension $d$.

**Experiments.** We performed an experimental evaluation of our algorithm on both synthetic and real data. Our evaluation allowed us to verify the accuracy and the sample complexity rates of our theoretical results. In all cases, the experiments validate the usefulness of our algorithm, which significantly outperforms previous approaches, almost exactly matching the best rate without noise.

**Related Work.** Question 1 fits in the framework of robust statistics [HR09, HRRS86]. Classical estimators from this field can be classified into two categories: either (i) they are computationally efficient but incur an error that scales *polynomially* with the dimension $d$, or (ii) they are provably robust (in the aforementioned sense) but are hard to compute. In particular, essentially all known estimators in robust statistics (e.g., the Tukey median [Tuk75]) have been shown [JP78, Ber06, HM13] to be intractable in the high-dimensional setting. We note that the robustness requirement does not typically pose information-theoretic impediments for the learning problem. In most cases of interest (see, e.g., [CGR15, CGR16, DKK+16]), the sample complexity of robust learning is comparable to its (easier) non-robust variant. The challenge is to design *computationally efficient* algorithms.

Efficient robust estimators are known for various *low-dimensional* structured distributions (see, e.g., [DDS14, CDSS13, CDSS14a, CDSS14b, ADLS16, ADLS17, DLS18]). However, the robust learning problem becomes surprisingly challenging in high dimensions. Recently, there has been algorithmic progress on this front: [DKK+16, LRV16] give polynomial-time algorithms with improved error guarantees for certain "simple" high-dimensional structured distributions. The results of [DKK+16] apply to simple distributions, including Bernoulli product distributions, Gaussians, and mixtures thereof (under some natural restrictions). Since the works of [DKK+16, LRV16], computationally efficient robust estimation in high dimensions has received considerable attention (see, e.g., [DKS17, DKK+17, BDLS17, DKK+18a, DKS18b, DKS18a, HL18, KSS18, PSBR18, DKK+18b, KKM18, DKS18c, LSLC18]).

## 1.2 Overview of Algorithmic Techniques

Our algorithmic approach builds on the framework of [DKK+16] with new technical and conceptual ideas. At a high level, our algorithm works as follows: We draw an $\epsilon$-corrupted set of samples from a

Bayesian network $P$ with known structure, and then iteratively remove samples until we can return the empirical conditional probability table.

First, we associate a vector $F(X)$ to each sample $X$ so that learning the mean of $F(X)$ to good accuracy is sufficient to recover the distribution. In the case of binary products, $F(X)$ is simply $X$, while in our case we need to take into account additional information about conditional means.

From this point, our algorithm will try to do one of two things: Either we show that the sample mean of $F(X)$ is close to the conditional mean of the true distribution (in which case we can already learn the ground-truth Bayes net $P$), or we are able to produce a *filter*, i.e., we can remove some of our samples, and it is guaranteed that we throw away more bad samples than good ones. If we produce a filter, we then iterate on those samples that pass the filter.

To produce a filter, we compute a matrix $M$ which is roughly the empirical covariance matrix of $F(X)$. We show that if the corruptions are sufficient to notably disrupt the sample mean of $F(X)$, there must be many erroneous samples that are all far from the mean in roughly the same direction, and we can detect this direction by looking at the largest eigenvector of $M$. If we project all samples onto this direction, concentration bounds of $F(X)$ will imply that almost all samples far from the mean are erroneous, and thus filtering them out will provide a cleaner set of samples.

**Organization.** Section 2 contains some technical results specific to Bayesian networks that we need. Section 3 gives the details of our algorithm and an overview of its analysis. In Section 4, we present the experimental evaluations. In Section 5, we conclude and propose directions for future work. Due to space constraints, we defer the proofs of the technical lemmas to the full version of the paper.

## 2   Technical Preliminaries

The structure of this section is as follows: First, we bound the total variation distance between two Bayes nets in terms of their conditional probability tables. Second, we define a function $F(x, q)$, which takes a sample $x$ and returns an $m$-dimensional vector that contains information about conditional means. Finally, we derive a concentration bound from Azuma's inequality.

**Lemma 5.** *Suppose that: (i) $\min_{k \in [m]} \Pr_P[\Pi_k] \geq \epsilon$, and (ii) $P$ or $Q$ is c-balanced, and (iii) $\frac{3}{c}\sqrt{\sum_k \Pr_P[\Pi_k](p_k - q_k)^2} \leq \epsilon$. Then we have that $d_{TV}(P, Q) \leq \epsilon$.*

Lemma 5 says that to learn a balanced fixed-structure Bayesian network, it is sufficient to learn all the relevant conditional means. However, each sample $x \sim P$ gives us information about $p_{i,a}$ only if $x \in \Pi_{i,a}$. To resolve this, we map each sample $x$ to an $m$-dimensional vector $F(x, q)$, and "fill in" the entries that correspond to conditional means for which the condition failed to happen. We will set these coordinates to their empirical conditional means $q$:

**Definition 6.** *Let $F(x, q)$ for $\{0,1\}^d \times \mathbb{R}^m \to \mathbb{R}^m$ be defined as follows: If $x \in \Pi_{i,a}$, then $F(x,q)_{i,a} = x_i$, otherwise $F(x,q)_{i,a} = q_{i,a}$.*

When $q = p$ (the true conditional means), the expectation of the $(i, a)$-th coordinate of $F(X, p)$, for $X \sim P$, is the same conditioned on either $\Pi_{i,a}$ or $\neg\Pi_{i,a}$. Using the conditional independence properties of Bayesian networks, we will show that the covariance of $F(x, p)$ is diagonal.

**Lemma 7.** *For $X \sim P$, we have $\mathbb{E}(F(X, p)) = p$. The covariance matrix of $F(X, p)$ satisfies $\text{Cov}[F(X, p)] = \text{diag}(\Pr_P[\Pi_k]p_k(1 - p_k))$.*

Our algorithm makes crucial use of Lemma 7 (in particular, that $\text{Cov}[F(X, p)]$ is diagonal) to detect whether or not the empirical conditional probability table of the noisy distribution is close to the conditional probabilities.

Finally, we will need a suitable concentration inequality that works under conditional independence properties. We can use Azuma's inequality to show that the projections of $F(X, q)$ on any direction $v$ is concentrated around the projection of the sample mean $q$.

**Lemma 8.** *For $X \sim P$, any unit vector $v \in \mathbb{R}^d$, and any $q \in [0, 1]^m$, we have*

$$\Pr[|v \cdot (F(X, q) - q)| \geq T + \|p - q\|_2] \leq 2\exp(-T^2/2) .$$

# 3 Robust Learning Algorithm

We first look into the major ingredients required for our filtering algorithm, and compare our proof with that for product distributions in [DKK$^+$16] on a more technical level.

In Section 2, we mapped each sample $X$ to $F(X, q)$ which contains information about the *conditional* means $q$, and we showed that it is sufficient to learn the mean of $F(X, q)$ to learn the ground-truth Bayes net.

Let $M$ denote the empirical covariance matrix of $(F(X, q) - q)$. We decompose $M$ into three parts: One coming from the ground-truth distribution, one coming from the subtractive error (because the adversary can remove $\epsilon N$ good samples), and one coming from the additive error (because the adversary can add $\epsilon N$ bad samples). We will make use of the following observations:

(1) The noise-free distribution has a diagonal covariance matrix.

(2) The term coming from the subtractive error has no large eigenvalues.

These two observations imply that any large eigenvalues of $M$ are due to the additive error. Finally, we will reuse our concentration bounds to show that if the additive errors are frequently far from the mean in a known direction, then they can be reliably distinguished from good samples.

For the case of binary product distributions in [DKK$^+$16], (1) is trivial because the coordinates are independent; but for Bayesian networks we need to expand the dimension of the samples and fill in the missing entries properly. Condition (2) is due to concentration bounds, and for product distributions it follows from standard Chernoff bounds, while for Bayes nets, we must instead rely on martingale arguments and Azuma's inequality.

## 3.1 Main Technical Lemma and Proof of Theorem 3

First, we need to show that a large enough set of samples with no noise satisfy properties we expect from a representative set of samples. We need that the mean, covariance, and tail bounds of $F(X, p)$ behave like we would expect them to. We call a set of samples that satisfies these properties $\epsilon$-good for $P$.

Our algorithm takes as input an $\epsilon$-corrupted multiset $S'$ of $N = \widetilde{\Omega}(m \log(1/\tau)/\epsilon^2)$ samples. We write $S' = (S \setminus L) \cup E$, where $S$ is the set of samples before corruption, $L$ contains the good samples that have been removed or (in later iterations) incorrectly rejected by filters, and $E$ represents the remaining corrupted samples. We assume that $S$ is $\epsilon$-good. In the beginning, we have $|E| + |L| \leq 2\epsilon|S|$. As we add filters in each iteration, $E$ gets smaller and $L$ gets larger. However, we will prove that our filter rejects more samples from $E$ than $S$, so $|E| + |L|$ must get smaller.

We will prove Theorem 3 by iteratively running the following efficient filtering procedure:

**Proposition 9** (Filtering). *Let $0 < \epsilon < 1/2$. Let $P$ be a $c$-balanced Bayesian network on $\{0, 1\}^d$ with known structure $G$. Assume each parental configuration of $P$ occurs with probability at least $\alpha \geq \Omega(\epsilon\sqrt{\log(1/\epsilon)}/c)$. Let $S' = S \cup E \setminus L$ be a set of samples such that $S$ is $\epsilon$-good for $P$ and $|E| + |L| \leq 2\epsilon|S'|$. There is an algorithm that, given $G$, $\epsilon$, and $S'$, runs in time $\widetilde{O}(d|S'|)$, and either*

*(i) Outputs a Bayesian network $Q$ with $d_{\mathrm{TV}}(P, Q) \leq \epsilon\sqrt{\ln(1/\epsilon)}/(c\alpha)$, or*

*(ii) Returns an $S'' = S \cup E' \setminus L'$ such that $|S''| \leq (1 - \frac{\epsilon}{d \ln d})|S'|$ and $|E'| + |L'| < |E| + |L|$.*

If this algorithm produces a subset $S''$, then we iterate using $S''$ in place of $S'$. We will present the algorithm establishing Proposition 9 in the following section. We first use it to prove Theorem 3.

**Proof of Theorem 3.** First a set $S$ of $N = \widetilde{\Omega}(m \log(1/\tau)/\epsilon^2)$ samples are drawn from $P$. We assume the set $S$ is $\epsilon$-good for $P$. Then an $\epsilon$-fraction of these samples are adversarially corrupted, giving a set $S' = S \cup E \setminus L$ with $|E|, |L| \leq \epsilon|S'|$. Thus $S'$ satisfies the conditions of Proposition 9, and the algorithm outputs a smaller set $S''$ of samples that also satisfies the conditions of the proposition, or else outputs a Bayesian network $Q$ with small $d_{\mathrm{TV}}(P, Q)$ that satisfies Theorem 3. Since $|S'|$ decreases if we produce a filter, eventually we must output a Bayesian network.

Next we analyze the running time. Observe that we can filter out at most $2\epsilon N$ samples, because we reject more bad samples than good ones. By Proposition 9, every time we produce a filter, we remove at least $\widetilde{\Omega}(d/\epsilon)|S'| = \widetilde{\Omega}(Nd/\epsilon)$ samples. Therefore, there are at most $\widetilde{O}(d)$ iterations, and each iteration takes time $\widetilde{O}(d|S'|) = \widetilde{O}(Nd)$ by Proposition 9, so the overall running time is $\widetilde{O}(Nd^2)$. $\square$

## 3.2 Algorithm `Filter-Known-Topology`

In this section, we present Algorithm 1 that establishes Proposition 9. [2]

---

**Algorithm 1** `Filter-Known-Topology`

---

1: **Input:** The dependency graph $G$ of $P$, $\epsilon > 0$, and a (possibly corrupted) set of samples $S'$ from $P$. $S'$ satisfies that there exists an $\epsilon$-good $S$ with $S' = S \cup E \setminus L$ and $|E| + |L| \le 2\epsilon|S'|$.
2: **Output:** A Bayes net $Q$ or a subset $S'' \subset S'$ that satisfies Proposition 9.
3: Compute the empirical conditional probabilities $q(i,a) = \Pr_{X \in_u S'}[X_i = 1 \mid \Pi_{i,a}]$.
4: Compute the empirical minimum parental configuration probability $\alpha = \min_{(i,a)} \Pr_{S'}[\Pi_{(i,a)}]$.
5: Define $F(X,q)$: If $x \in \Pi_{i,a}$ then $F(x,q)_{i,a} = x_i$, otherwise $F(x,q)_{i,a} = q_{i,a}$ (Definition 6).
6: Compute the empirical second-moment matrix of $F(X,q) - q$ and zero its diagonal, i.e., $M \in \mathbb{R}^{m \times m}$ with $M_{k,k} = 0$, and $M_{k,\ell} = \mathbb{E}_{X \in_u S'}[(F(X,q)_k - q_k)(F(X,q)_\ell - q_\ell)^T]$ for $k \ne \ell$.
7: Compute the largest (in absolute value) eigenvalue $\lambda^*$ of $M$, and the associated eigenvector $v^*$.
8: **if** $|\lambda^*| \le O(\epsilon \log(1/\epsilon)/\alpha)$ **then**
9:     Return $Q = $ the Bayes net with graph $G$ and conditional probabilities $q$.
10: **else**
11:     Let $\delta := 3\sqrt{\epsilon|\lambda^*|}/\alpha$. Pick any $T > 0$ that satisfies

$$\Pr_{X \in_u S'}[|v^* \cdot (F(X,q) - q)| > T + \delta] > 7\exp(-T^2/2) + 3\epsilon^2/(T^2 \ln d).$$

    Return $S'' = $ the set of samples $x \in S'$ with $|v \cdot (F(x,q) - q)| \le T + \delta$.

---

At a high level, Algorithm 1 computes a matrix $M$, and shows that: either $\|M\|_2$ is small, and we can output the empirical conditional probabilities, or $\|M\|_2$ is large, and we can use the top eigenvector of $M$ to remove bad samples.

**Setup and Structural Lemmas.** In order to understand the second-moment matrix with zeros on the diagonal, $M$, we will need to break down this matrix in terms of several related matrices, where the expectation is taken over different sets. For a set $D = S', S, E$ or $L$, we use $w_D = |D|/|S'|$ to denote the fraction of the samples in $D$. Moreover, we use $M_D = \mathbb{E}_{X \in_u D}[((F(X,q) - q)(F(X,q) - q)^T]$ to denote the second-moment matrix of samples in $D$, and let $M_{D,0}$ be the matrix we get from zeroing out the diagonals of $M_D$. Under this notation, we have $M_{S'} = w_S M_S + w_E M_E - w_L M_L$ and $M = M_{S',0}$.

Our first step is to analyze the spectrum of $M$, and in particular show that $M$ is close in spectral norm to $w_E M_E$. To do this, we begin by showing that the spectral norm of $M_{S,0}$ is relatively small. Since $S$ is good, we have bounds on the second moments $F(X,p)$. We just need to deal with the error from replacing $p$ with $q$:

**Lemma 10.** $\|M_{S,0}\|_2 \le O(\epsilon + \sqrt{\sum_k \Pr_S[\Pi_k](p_k - q_k)^2} + \sum_k \Pr_S[\Pi_k](p_k - q_k)^2)$.

Next, we wish to bound the contribution to $M$ coming from the subtractive error. We show that this is small due to concentration bounds on $P$ and hence on $S$. The idea is that for any unit vector $v$, we have tail bounds for the random variable $v \cdot (F(X,q) - q)$ and, since $L$ is a subset of $S$, $L$ can at worst consist of a small fraction of the tail of this distribution.

**Lemma 11.** $w_L \|M_L\|_2 \le O(\epsilon \log(1/\epsilon) + \epsilon\|p - q\|_2^2)$.

Finally, combining the above results, since $M_S$ and $M_L$ have small contribution to the spectral norm of $M$ when $\|p - q\|_2$ is small, most of it must come from $M_E$.

**Lemma 12.**
$\|M - w_E M_E\|_2 \le O\left(\epsilon \log(1/\epsilon) + \sqrt{\sum_k \Pr_{S'}[\Pi_k](p_k - q_k)^2} + \sum_k \Pr_{S'}[\Pi_k](p_k - q_k)^2\right).$

Lemma 12 follows using the identity $|S'|M = |S|M_{S,0} + |E|M_{E,0} - |L|M_{L,0}$ and bounding the errors due to the diagonals of $M_E$ and $M_L$.

**The Case of Small Spectral Norm.** In this section, we will prove that if $\|M\|_2 = O(\epsilon \log(1/\epsilon)/\alpha)$, then we can output the empirical conditional means $q$. Recall that $M_{S'} = \mathbb{E}_{X \in_u S'}[(F(X,q)_i - q_i)(F(X,q)_j - q_j)^T]$ and $M = M_{S',0}$.

We first show that the contributions that $L$ and $E$ make to $\mathbb{E}_{X \in_u S'}[F(X,q) - q]$ can be bounded in terms of the spectral norms of $M_L$ and $M_E$. It follows from the Cauchy-Schwarz inequality that:

**Lemma 13.** $\|\mathbb{E}_{X \in_u L}[F(X,q) - q]\|_2 \leq \sqrt{\|M_L\|_2}$ and $\|\mathbb{E}_{X \in_u E}[F(X,q) - q]\|_2 \leq \sqrt{\|M_E\|_2}$.

Combining with the results about these norms in Section 3.2, Lemma 13 implies that if $\|M\|_2$ is small, then $q = \mathbb{E}_{X \in_u S'}[F(X,q)]$ is close to $\mathbb{E}_{X \in_u S}[F(X,q)]$, which is then necessarily close to $\mathbb{E}_{X \sim P}[F(X,p)] = p$. The following lemma states that the mean of $(F(X,q) - q)$ under the good samples is close to $(p - q)$ scaled by the probabilities of parental configurations under $S'$:

**Lemma 14.** Let $z \in \mathbb{R}^m$ be the vector with $z_k = \Pr_{S'}[\Pi_k](p_k - q_k)$. Then $\|\mathbb{E}_{X \in_u S}[F(X,q) - q] - z\|_2 \leq O(\epsilon(1 + \|p - q\|_2))$.

Note that $z$ is closely related to the total variation distance between $P$ and $Q$ (the Bayes net with conditional probabilities $q$). We can write $(\mathbb{E}_{X \in_u S'}[F(X,q)] - q)$ in terms of this expectation under $S$, $E$, and $L$ whose distance from $q$ can be upper bounded using the previous lemmas. Using Lemmas 11, 12, 13, and 14, we can bound $\|z\|_2$ in terms of $\|M\|_2$:

**Lemma 15.** $\sqrt{\sum_k \Pr_{S'}[\Pi_k]^2(p_k - q_k)^2} \leq 2\sqrt{\epsilon\|M\|_2} + O(\epsilon\sqrt{\log(1/\epsilon)} + 1/\alpha)$.

Lemma 15 implies that, if $\|M\|_2$ is small then so is $\sqrt{\sum_k \Pr_{S'}[\Pi_k]^2(p_k - q_k)^2}$. We can then use it to show that $\sqrt{\sum_k \Pr_P[\Pi_k](p_k - q_k)^2}$ is small. We can do so by losing a factor of $1/\sqrt{\alpha}$ to remove the square on $\Pr_{S'}[\Pi_k]$, and showing that $\min_k \Pr'_S[\Pi_k] = \Theta(\min_k \Pr_P[\Pi_k])$ when it is at least a large multiple of $\epsilon$. Finally, if $\sqrt{\sum_k \Pr_P[\Pi_k](p_k - q_k)^2}$ is small, Lemma 5 tells us that $d_{TV}(P, Q)$ is small. This completes the proof of the first case of Proposition 9.

**Corollary 16** (Part *(i)* of Proposition 9). *If* $\|M\|_2 \leq O(\epsilon \log(1/\epsilon)/\alpha)$, *then* $d_{TV}(P, Q) = O(\epsilon\sqrt{\log(1/\epsilon)}/(c\min_k \Pr_P[\Pi_k]))$.

**The Case of Large Spectral Norm.** Now we consider the case when $\|M\|_2 \geq C\epsilon \ln(1/\epsilon)/\alpha$. We begin by showing that $p$ and $q$ are not too far apart from each other. The bound given by Lemma 15 is now dominated by the $\|M\|_2$ term. Lower bounding the $\Pr_{S'}[\Pi_k]$ by $\alpha$ gives the following claim.

**Claim 17.** $\|p - q\|_2 \leq \delta := 3\sqrt{\epsilon\|M\|_2}/\alpha$.

Recall that $v^*$ is the largest eigenvector of $M$. We project all the points $F(X,q)$ onto the direction of $v^*$. Next we show that most of the variance of $(v^* \cdot (F(X,q) - q))$ comes from $E$.

**Claim 18.** $v^{*T}(w_E M_E)v^* \geq \frac{1}{2}v^{*T}Mv^*$.

Claim 18 follows from the observation that $\|M - w_E M_E\|_2$ is much smaller than $\|M\|_2$. This is obtained by substituting the bound on $\|p - q\|_2$ (in terms of $\|M\|_2$) from Claim 17 into the bound on $\|M - w_E M_E\|_2$ given by Lemma 12.

Claim 18 implies that the tails of $w_E E$ are reasonably thick. In particular, we show that there must be a threshold $T > 0$ satisfying the desired property in Step 9 of our algorithm.

**Lemma 19.** *There exists a* $T \geq 0$ *such that*

$$\Pr_{X \in_u S'}[|v \cdot (F(X,q) - q)| > T + \delta] > 7\exp(-T^2/2) + 3\epsilon/(T^2 \ln d).$$

If Lemma 19 were not true, by integrating this tail bound, we can show that $v^{*T}M_E v^*$ would be small. Therefore, Step 11 of Algorithm 11 is guaranteed to find some valid threshold $T > 0$.

Finally, we show that the set of samples $S''$ we return after the filter is better than $S'$ in terms of $|L| + |E|$. This completes the proof of the second case of Proposition 9.

**Claim 20** (Part *(ii)* of Proposition 9). *If we write* $S'' = S \cup E' \setminus L'$, *then* $|E'| + |L'| < |E| + |L|$ *and* $|S''| \leq (1 - \frac{\epsilon}{d \ln d})|S'|$.

Claim 9 follows from the fact that $S$ is $\epsilon$-good, so we only remove at most $(3\exp(T^2/2) + \epsilon/T^2\log d)|S|$ samples from $S$. Since we remove more than twice as many samples from $S'$, most of the samples we throw away are from $E$. Moreover, we remove at least $(1 - \frac{\epsilon}{d\ln d})|S'|$ samples because we can show that the threshold $T$ is at most $\sqrt{d}$.

**Running Time of Our Algorithm 1**  First, $q$ and $\alpha$ can be computed in time $O(Nd)$ because each sample only affects $d$ entries of $q$. We do not explicitly write down $F(X, q)$ or $M$. Then, we use the power method to compute the largest eigenvalue $\lambda^*$ of $M$ and the associated eigenvector $v^*$. In each iteration, we implement matrix-vector multiplication with $M$ by writing $Mv$ as $\sum_i((F(x_i, q) - q)^T v)(F(x_i, q) - q)$ for any vector $v \in \mathbb{R}^m$. Because each $(F(x_i, q) - q)$ is $d$-sparse, computing $Mv$ takes time $O(dN)$. The power method takes $(\log m/\epsilon')$ iterations to find a $(1 - \epsilon')$-approximately largest eigenvalue. We can set $\epsilon'$ to a small constant, because we can tolerate a small multiplicative error in estimating the spectral norm of $M$ and we only need an approximate top eigenvector (see, e.g., Corollary 16 and Lemma 18). Thus, the power method takes time $O(dN \log m)$. Finally, computing $|v^* \cdot (F(x, q) - q)|$ takes time $O(dN)$, then we can sort the samples and find a threshold $T$ in time $O(N \log N)$, and throw out the samples in time $O(N)$.

# 4  Experiments

We test our algorithms using data generated from both synthetic and real-world networks (e.g., the ALARM network [BSCC89]) with synthetic noise. All experiments were run on a laptop with 2.6 GHz CPU and 8 GB of RAM. We found that our algorithm achieves the smallest error consistently in all trials, and that the error of our algorithm almost matches the error of the empirical conditional probabilities of the uncorrupted samples. Moreover, our algorithm can easily scale to thousands of dimensions with millions of samples. [3]

## 4.1  Synthetic Experiments

The results of our synthetic experiments are shown in Figure 1. In the synthetic experiment, we set $\epsilon = 0.1$ and first generate a Bayes net $P$ with $100 \leq m \leq 1000$ parameters. We then generate $N = \frac{10m}{\epsilon^2}$ samples, where a $(1 - \epsilon)$-fraction of the samples come from the ground truth $P$, and the remaining $\epsilon$-fraction come from a noise distribution. The goal is to output a Bayes net $Q$ that minimizes $d_{\mathrm{TV}}(P, Q)$.

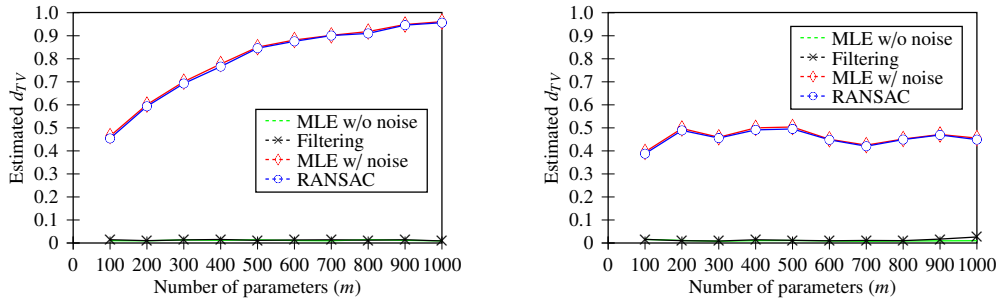

Figure 1: Experiments with synthetic data: error is reported against the size of the conditional probability table (lower is better). The error is the estimated total variation distance to the ground truth Bayes net. We use the error of MLE without noise as our benchmark. We plot the performance of our algorithm (`Filtering`), empirical mean with noise (`MLE`), and `RANSAC`. We report two settings: the underlying structure of the Bayes net is a random tree (left) or a random graph (right).

We draw the parameters of $P$ independently from $[0, 1/4] \cup [3/4, 1]$ uniformly at random, i.e., in a setting where the "balancedness" assumption does not hold. Our experiments show that our filtering algorithm works very well in this setting, even when the assumptions under which we can prove theoretical guarantees are not satisfied. This complements our theoretical results and illustrates that our algorithm is not limited by these assumptions and can apply to more general settings in practice.

In Figure 1, we compare the performance of (1) our filtering algorithm, (2) the empirical conditional probability table with noise, and (3) a RANSAC-based algorithm (see the end of Section 4 for a detailed description). We use the error of the empirical conditional mean without noise (i.e., MLE estimator with only good samples) as the gold standard, since this is the best one could hope for even if all the corrupted samples are identified. We tried various graph structures for the Bayes net $P$ and noise distributions, and similar patterns arise for all of them. In the top figure, the dependency graph of $P$ is a randomly generated tree, and the noise distribution is a binary product distribution; In the bottom figure, the dependency graph of $P$ is a random graph, and the noise distribution is the tree Bayes net used as the ground truth in the first experiment.

## 4.2 Semi-Synthetic Experiments

In the semi-synthetic experiments, we apply our algorithm to robustly learn real-world Bayesian networks. The ALARM network [BSCC89] is a classic Bayes net that implements a medical diagnostic system for patient monitoring.

Our experimental setup is as follows: The underlying graph of ALARM has 37 nodes and 509 parameters. Since the variables in ALARM can have up to 4 different values, we first transform it into an equivalent binary-valued Bayes net. After the transformation, the network has $d = 61$ nodes and $m = 820$ parameters. We are interested in whether our filtering algorithm can learn a Bayes net that is "close" to ALARM when samples are corrupted; and how many corrupted samples can our algorithm tolerate. For $\epsilon = [0.05, 0.1, \ldots, 0.4]$, we draw $N = 10^6$ samples, where a $(1 - \epsilon)$-fraction of the samples come from ALARM, and the other $\epsilon$-fraction comes from a noise distribution.

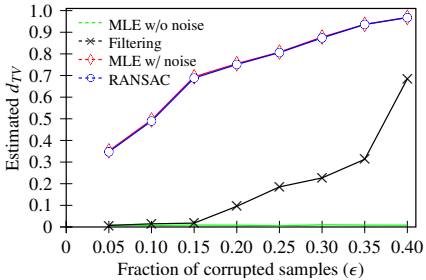

Figure 2: Experiments with semi-synthetic data: error is reported against the fraction of corrupted samples (lower is better). The error is the estimated total variation distance to the ALARM network. We use the sampling error without noise as a benchmark, and compare the performance of our algorithm (`Filtering`), empirical mean with noise (`MLE`), and `RANSAC`.

In Figure 2, we compare the performance of (1) our filtering algorithm, (2) the empirical conditional means with noise, and (3) a RANSAC-based algorithm. We use the error of the empirical conditional means without noise as the gold standard. We tried various noise distributions and observed similar patterns. In Figure 2, the noise distribution is a Bayes net with random dependency graphs and conditional probabilities drawn from $[0, \frac{1}{4}] \cup [\frac{3}{4}, 1]$ (same as the ground-truth Bayes net in Figure 1).

The experiments show that our filtering algorithm outperforms `MLE` and `RANSAC`, and that the error of our algorithm degrades gracefully as $\epsilon$ increases. It is worth noting that even the ALARM network does not satisfy our balancedness assumption on the parameters, our algorithm still performs well on it and recovers the conditional probability table of ALARM in the presence of corrupted samples.

## 5 Conclusions and Future Directions

In this paper, we initiated the study of the efficient robust learning for graphical models. We described a computationally efficient algorithm for robustly learning Bayesian networks with a known topology, under some mild assumptions on the conditional probability table. We evaluate our algorithm experimentally, and we view our experiments as a proof of concept demonstration that our techniques can be practical for learning fixed-structure Bayesian networks. A challenging open problem is to generalize our results to the case when the underlying directed graph is unknown.

This work is part of a broader agenda of systematically investigating the robust learnability of high-dimensional structured probability distributions. There is a wealth of natural probabilistic models that merit investigation in the robust setting, including undirected graphical models (e.g., Ising models), and graphical models with hidden variables (i.e., incorporating latent structure).

**Acknowledgements.** We are grateful to Daniel Hsu for suggesting the model of Bayes nets, and for pointing us to [Das97]. Yu Cheng is supported in part by NSF CCF-1527084, CCF-1535972, CCF-1637397, CCF-1704656, IIS-1447554, and NSF CAREER Award CCF-1750140. Ilias Diakonikolas is supported by NSF CAREER Award CCF-1652862 and a Sloan Research Fellowship. Daniel Kane is supported by NSF CAREER Award CCF-1553288 and a Sloan Research Fellowship.

## Footnotes

[1] Throughout the paper, we use $\widetilde{O}(f)$ to denote $O(f\,\mathrm{polylog}(f))$.

[2] We use $X \in_u S$ to denote that the point $X$ is drawn uniformly from the set of samples $S$.

[3] The bottleneck of our algorithm is fitting millions of samples of thousands dimension all in the memory.

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
