[Reviews · NeurIPS 2018]

Reviewer 1



This paper is on the robust learning of Bayesian networks from data. In particular, we assume the structure is known, and that we want to estimate the parameters from a dataset where a fraction of the examples have been adversarially corrupted (they come from some other distribution). The authors provide an algorithm that is both computationally and statistically efficient. They also provide experiments verifying their approach, and further demonstrating the usefulness of their algorithm even when the network to be learned is outside of the scope of the theoretical assumptions made for the bounds. The paper is relatively technical, but the authors do a good job explaining the higher-level ideas and intuitions. The experiments were also nice, demonstrating that the algorithm works well in practice, even in situations that it was not designed for.

Reviewer 2



I preface this by saying that I have reviewed this paper once for NIPS 2016, and re-read it. It seems the paper has no essential changes, so my opinion is largely the same. The paper considers the problem of learning the parameters of a Bayes net with known structure, given samples from it with potentially adversarial noise. The main goal is to get bounds on the samples that are independent of dimension. The main requirements on the Bayes net parameters are reasonable: the probability of any configuration of the parents is reasonable and the conditional probabilities on any edge are bounded away from 0 and 1. The algorithm is quite similar morally to [DKK+16] -- on a high level it designs a quantity F(X,q), s.t. E[F(X,p)] = p, where p is the conditional probability "table" specified by the Bayes net and q is the empirical conditional probability table. The covariance matrix of F(X,q) on the other hand, "catches" the adversarial noise: if the covariance has a large eigenvalue, one can "filter" the "culprit" samples, and repeat the process. Otherwise, the empirical estimate is close in TV distance. The paper is well-written, though the proofs are straightforward and natural (and largely follow the structure of [DKK+16]). There are some new tricks (e.g. how one "fills in" F(X,q) in the unobserved spots; why the mean of F(X,p) is diagonal; and how to use the spikes in the spectrum to produce a filter), but I feel like it's a poor fit for NIPS for two reasons. For one, the techniques are quite similar to [DKK+16]; For another, it may be of interest to a relatively specialized crowd, both in terms of theoretical tools and practical algorithms. As mentioned a previous time I reviewed this paper: the more interesting question is structure-learning Bayes nets, even under "benign" noise assumptions. _______________________ Edit after rebuttal: after the rebuttal/discussions with other reviewers, my opinion is largely the same, though I may have scored a bit harshly. Correspondingly, I raised my score up by a point.

Reviewer 3



The authors present an algorithm for estimating parameters in a Bayesian network over binary variables with *known structure* in the presence of adversarial noise. The algorithm is shown to achieve near-optimal performance in polynomial time. The paper appears to be a solid theoretical piece. I haven't checked any of the details. The interest to the NIPS community is probably limited but I cannot identify other significant drawbacks. The performance of the algorithm is demonstrated on simulated data and it outperforms the naive estimate (just use all data) and a RANSAC-based technique. (The authors state that the RANSAC-technique is described at the end of Sec. 4 but it isn't.)